

# The temperature-dependent shear strength of ice-filled joints in rock mass considering the effect of joint roughness, opening and shear rates

Shibing Huang[1,2], Haowei Cai[1], Zekun Xin[1], Gang Liu[1]

[1] School of Resources and Environmental Engineering, Wuhan University of Science and Technology, Wuhan, Hubei 430081, China

[2] Hubei Key Laboratory for Efficient Utilization and Agglomeration of Metallurgic Mineral Resources, Wuhan University of Science and Technology, Wuhan, Hubei 430081, China

*Correspondence to*: Shibing Huang (huangshibing@wust.edu.cn)

**Abstract.** Global warming causes many rockfall activities of the alpine mountains, especially when ice-filled joints in the rock mass become thawed. The warming and thawing of frozen soils and intact rocks was widely studied in the past several decades, however, the variation of shear strengths of ice-filled joints was not fully understood. Recently, some scholars studied the thawing process and strength loss of ice-filled joints at different temperatures, however, the influence of the joint roughness, opening and shear rate on ice-filled joints was not systematically investigated. In this study, a series of compression-shear experiments were conducted to investigate the shear strength of ice-filled rock joints by considering the effects of joint roughness, opening and shear rates. The shear strength quickly reduces with increasing temperature, especially above -1 ℃. In addition, the shear strength decreases with increasing joint openings but it increases with increasing joint roughness. When the joint opening is large enough, the effect of joint roughness disappears and the shear strength of ice-filled joints is equal to that of solid ice.





Increasing shear rate will decrease the shear strength of ice-filled joints because the joint ice displays the
brittle failure phenomenon at a high shear rate. The Mohr-coulomb criterion also can be used to
characterize the relationship between the shear strength and the normal stress of ice-filled joints.
However, a general strength model by considering the joint opening, normal stress and joint roughness
should be proposed by a further study. This research can provide a better understanding of the warming
degradation mechanism of ice-filled joints by considering the above important influencing factors.
**1 Introduction**
With the increase of global temperature and human activities in permafrost areas, many alpine rock
masses become more unstable (Gruber and Haeberli, 2007; Allen and Huggle, 2013; Hartmeyer et al.,
2020; Legay et al., 2021; Hilger et al., 2021). A large number of rockfalls in permafrost alpine bedrock
slopes indicated the exposure of broken ice after shear failure, which could cause serious natural
geological disasters (Krautblatter et al., 2021; Walter et al., 2019). For example, the rockfall disaster that
happened in Chamoli, Indian Himalaya, in 2021 took more than 200 lives and destroyed two hydropower
facilities (Shugar et al., 2021). According to investigation results, this rockfall disaster was caused by the
warming and thawing of ice. Although the freezing expansion process of joint ice was harmful for the
stability of joint rock masses, the bonding strength between ice and joint wall can strengthen the joints
themselves after complete freezing (Matsuoka and Murton, 2008; Zhang et al., 2020; Shan et al., 2021).
However, if the joint ice was thawed, the rock-ice-rock "sandwich" structure would be debonded and
unstable. In addition, the liquid water produced by warming ice could lower the friction between joint
surface and thus reduced the stability of joint rock slopes (Zhao et al., 2017). Many field data showed
that most of the irreversible fracture displacement and rockfall happened in the warm seasons instead of



the cool seasons because the warming and thawing of joint ice could greatly decrease the strength of rock
mass containing ice-filled joints (Weber et al., 2018; Etzelmüller et al., 2022). Yang et al. (2019) claimed
that the existence of detached ice block could promote the mobility of ice-rock system and thus cause a
more serious geological disaster on alpine rock slope. Therefore, the warming degradation of the ice-
rock interface and the strength loss of ice-filled joints should be comprehensively studied.
In the past decades, the warming degradation of permafrost soils was widely investigated, however, there
is little literature reporting the strength loss of rocks containing ice-filled joints. The shear experiment of
the ice-rock interface might be first conducted by replacing the rock with concrete in order to make a
specific roughness (Davies et al., 2001, 2017). These experiments were conducted at the temperature
from -5 to 0 ℃. Krautblatter et al. (2012) developed a shear strength model for the ice-filled joints that
incorporates the cracking of rock bridges, the friction of rough joint walls, creep of ice and detachment
of rock-ice interfaces. Mamot et al. (2018) conducted a systematic study of the shear failure of limestone-
ice and mica-rich interfaces at constant strain rates from -10 to -0.5℃, and they found that the normal
stress and freezing temperature were two important factors influencing the shear strength. However, the
uniform joint surfaces were used without considering the influence of joint roughness. Mamot et al. (2021)
further predicted the warming stability of permafrost slopes containing ice-filled joints by using the
Universal Distinct Element Code (UDEC). The simulation results verified that the warming temperature
close to the melting point might drive the slide of a slope with angle of 50°-62°, and the actual slope
angle also depended on the joint orientation. The above research mainly investigated the thawing
temperature and normal stress on the shear strength of ice-filled joints. The highest normal stress is about
1.438 MPa (Davies et al., 2001), and the maximum range for the temperature was -10 ℃ to -0.5 ℃





(Mamot et al., 2018). However, the freezing depth could exceed 100 m for some alpine caves containing
frozen ice (normal stress large than 2 MPa) and the temperature was less than -15 ℃ as observed in the
field (Colucci and Guglielmin, 2019). Therefore, a much wider range of temperature and normal stress
should be considered when investigating the shear characteristics of ice-filled joints.
In addition, although some scholars began to pay attention to the mechanical properties of ice-filled joint
rock mass, the influence of many important factors on the shear strength of ice-filled joints was not
investigated, including the joint roughness, shear rate, normal stress and joint opening. Generally, the
natural joints have different roughness and openings (Shen et al., 2020). In this study, a comprehensive
shear experiment was performed on the ice-filled joints in sandstones. The main purpose was to reveal
the influencing mechanism of freezing temperature, joint roughness, shear rate, joint opening and normal
stress on the shear strength of ice-filled joints in rock masses. This research can provide a better
understanding of the warming degradation process of the ice-filled joints and the thawing disaster of
alpine mountains in cold regions.
**2 Materials and methods**
**2.1 Collection of sandstones**
The red sandstones collected from Yichang city of Hubei province were used in this experiment. This is
a typical sedimentary rock and is widely distributed on the surface of the earth. The block samples with
approximately equal P-wave (compressional wave) velocities were chosen to make frozen samples
containing ice-filled joints. The basic physico-mechanical properties of this red sandstone are given in
Table 1.





**Table 1.** The basic physico-mechanical properties of the fresh sandstone. $\rho$: density. $n$: porosity. $V_p$: primary.
wave velocity. $\tau_{ps}$: shear strength. UCS: uniaxial compressive strength.

| $\rho$ (g/cm³) | $n$ (%) | $V_p$ (m/s) | | $\tau_{ps}$ (MPa) | | UCS (MPa) | |
|---|---|---|---|---|---|---|---|
| | | Dry | Saturated | Dry | Saturated | Dry | Saturated |
| 2.32 | 7.71 | 2992 | 3264 | 7.60 | 3.02 | 79.53 | 30.97 |


**2.2 Preparation of ice-filled joint rock mass**
According to the JRC index proposed by Barton and Choubey (1977), five kinds of roughness were used
in this experiment, including No. 2 (2°-4°), No. 4 (6°-8°), No. 6 (10°-12°), No. 8 (14°-16°) and No. 10
(18°-20°), respectively. The frozen samples containing ice-filled joints are made in the laboratory
because it is hard to cut or drill them in the fields. The manufacturing process of ice-filled joint rock mass
mainly includes the following steps:
①    The original rock blocks were cut into the designed rectangular blocks (100 mm × 100 mm × 50
mm) by using a rock cutting machine.
②    These rectangular blocks were used to engrave different rough curves on the surface by using a 3D
numerical control engraving machine. The roughness can be controlled by implanting the standard JRC
curves into the controlling system of this machine. Each frozen rock sample containing an ice-filled joint
was assembled by using a pair of rectangular blocks with the same roughness.
③    The rock blocks were heated in a dry oven at 105 ℃ in order to tightly paste the waterproof tape
and prevent the escape of joint water during freezing.



④   The joint opening was divided into different specified thicknesses which were controlled by
inserting rubber strips, and a piece of waterproof tape was pasted on the surface in order to store water.
⑤   When the waterproof tape was tightly bonded on the rock surface, liquid water should be injected
into the artificial joint until no water leaks out. After that, the water-filled joint rock mass was put into a
steel mold to freeze in a freezing chamber. The steel mold was used to control the joint opening because
the volume of joint water would expand during freezing. Then ice-filled joint samples can be derived
after freezing at -20 ℃ for 12 h. The manufacturing procedure and related ice-filled joint samples were
shown in Fig. 1.
**Table 2.** Ten standard joint profiles (Barton and Choubey, 1977).

| Profile No. | Typical roughness profiles | JRC range |
|---|---|---|
| No. 1 | | 0-2(0.4) |
| No. 2 | | 2-4(2.8) |
| No. 3 | | 4-6(5.8) |
| No. 4 | | 6-8(6.7) |
| No. 5 | | 8-10(9.5) |
| No. 6 | | 10-12(10.8) |
| No. 7 | | 12-14(12.8) |
| No. 8 | | 14-16(14.5) |



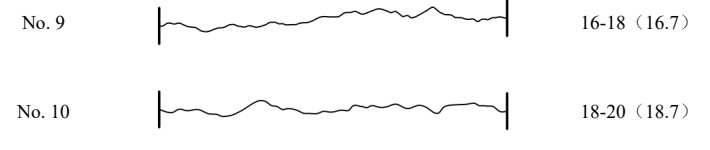


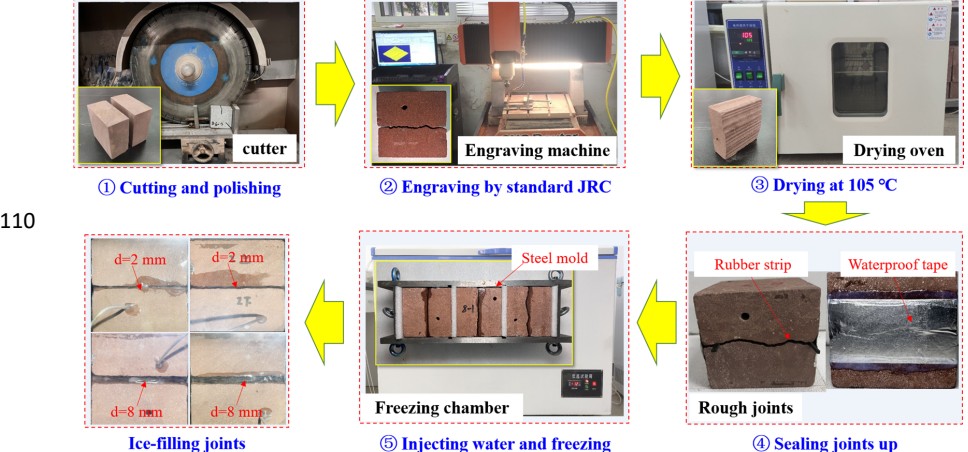


**Figure 1.**  Preparation of ice-filled joints. The preparation steps are as follows: ① Cutting and polishing, ②
Engraving by standard, ③ Drying at 105 ℃, ④ Sealing joints up, ⑤ Injecting water and freezing.
**2.3 Experimental procedures**
The main objective of this study is to investigate the effect of critical factors on the shear strength of ice-
filled joint rock mass, including the freezing temperature, joint roughness, shear rates, joint opening and
normal stress. The joint roughness is a basic index for rock joints, which is always considered when
investigating other factors. Therefore, all the samples can be divided into 4 groups, namely the
temperature group, shear rate group, joint opening group, and normal stress group. In the pre-test, the
shear strength of the ice-filled joint does not change when the temperature is below -5 ℃, however, it
greatly decreases when the temperature increases from -5 ℃ to 0 ℃. Therefore, the temperatures are set



as -15 ℃, -5 ℃, -1 ℃ and -0.5 ℃, respectively. The shear rates are 0.2 mm/min, 0.4mm/min and
0.8mm/min in the shear rate group. In the joint opening group, the openings of ice-filled joints are 2 mm,
8 mm and 14 mm, respectively. The freezing depth on the earth may be small, however, it can exceed
100 m in some alpine caves, where the in-situ stress is close to 2 MPa. Therefore, in the normal stress
group, the normal stresses are set as 0 MPa, 0.5 MPa, 1 MPa, 1.5 MPa and 2 MPa, respectively. Three
parallel experiments were performed on each group to eliminate the discreteness of ice-filled joint
samples and experiment error. There are approximately 225 ice-filled joint samples prepared in this
experiment. The distribution of these ice-filled joint samples were shown in Fig. 2.
All the water-containing joints were frozen in a freeze box at a specific temperature for about 12 h, and
they were used to conduct the direct shear experiment on a temperature-controlled shearing instrument
under the scheduled low temperature and normal stress. A temperature sensor was implanted into the
sample to accurately monitor the internal temperature change of ice-filled joint samples. When the
scheduled freezing temperature was reached, the normal stress was applied with a loading rate of 0.2
kN/s. Then the shear process was performed in the displacement mode with the designed shear rate. After
the shear experiment, the rupture modes of ice-filled joints were captured and analyzed by using a camera.



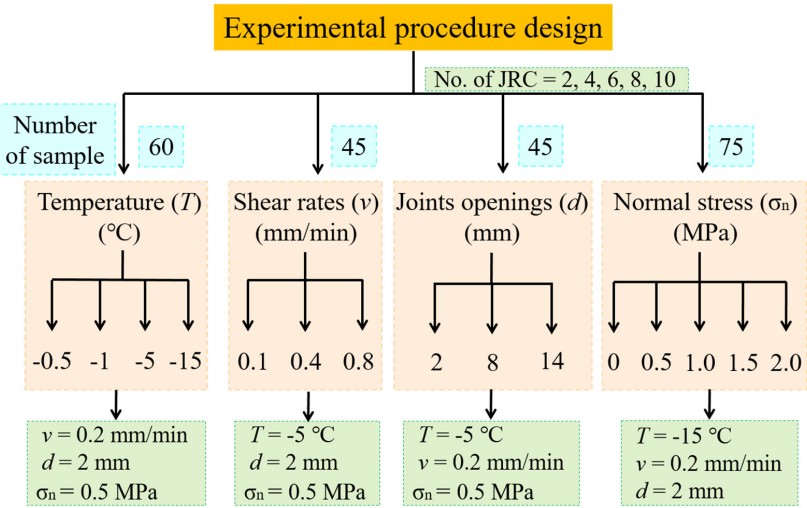


**Figure 2.** Distribution of rock samples containing ice-filled joints. *T*: Temperature. *v*: Shear rates. *d*: Joint openings.

$\sigma_n$: Normal stress.

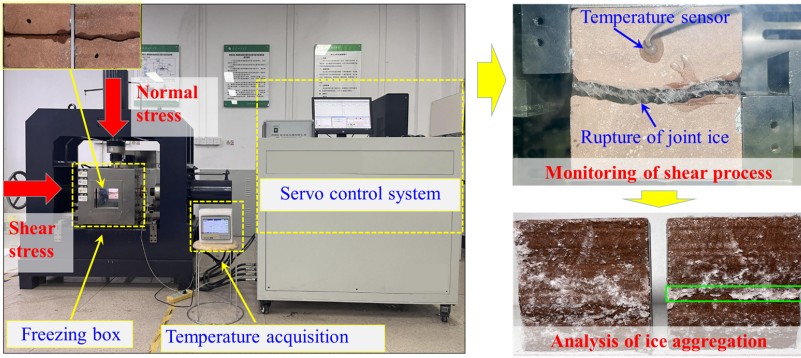


**Figure 3.** Shear experiment procedure and equipment



## 3 Experimental results

### 3.1 Effect of freezing temperature and joint roughness

In the temperature group, freezing temperatures were set as -15 ℃, -5 ℃, -1 ℃ and -0.5 ℃, and the joint roughness was named by the profile number in Table 2. The shear strength is dependent on the freezing temperature and joint roughness as shown in Fig. 4. The shear strength decreases remarkably with increasing freezing temperature. When the temperature increases from -15 ℃ to -0.5 ℃, the mean strength decreases by approximately 54%, 32%, 60%, 46% and 56% for profiles of No. 2, No. 4, No. 6, No. 8 and No. 10, respectively. The shear strength of ice-filled joints does not always increase with JRC, which has a considerable reduction at the joint profiles of No. 6 and No. 10. It illustrates that solid ice is a kind of special infilling material, which is different from soft soils or cement-based materials. The change law of shear strength against JRC may be explained by the shear rupture mode as shown in Fig. 5. There are several aggregation regions of rupture ice close to large climbing bulges on the surface of joints. The peak shear strength of ice-filled joints is related to the aggregation area of rupture ice; because a large shear force is required to promote the solid ice to shear slide along the slope of bulges. The aggregation area and location along the rough profile of joints after shear failure are plotted in Fig. 6. It can be observed that the aggregation ice appears before several high bulges and the aggregation location is almost independent of the freezing temperature if aggregation ice occurs. The climbing bulges in front of the aggregation ice are noticeable and influential. It implies that the influence of joint roughness on the shear strengths of these ice-filled joints may be only controlled by several noticeable bulges instead of the JRC index. Figure 7 shows that the shear strengths of No. 6 and No. 10 display obvious reduction trends, which may be in accordance with the ice aggregation area. The ice aggregation area decreases




with increasing the freezing temperature, because the bonding strength between ice and joint surface
becomes to be weaker, and the shear rupture happens along the ice-rock interface instead of solid ice
when the freezing temperature is larger than -0.5 ℃.
In addition, when the freezing temperature is close to 0 ℃, the pre-melting of ice-rock interface induced
by the normal stress will cause a reduction of bonding strength. Therefore, the shear strength between
bonded ice-rock interfaces is much smaller than the shear strength of solid ice at a high freezing
temperature close to the melting point of bulk ice, such as -0.5 ℃. It should be noted that the aggregation
phenomenon of rupture ice disappears when $T$ = -0.5 ℃ because the high-temperature ice is ductile failure
along the ice-rock interface instead of the joint ice itself. However, the climbing effect still makes a
significant contribution to the increase of shear strength.

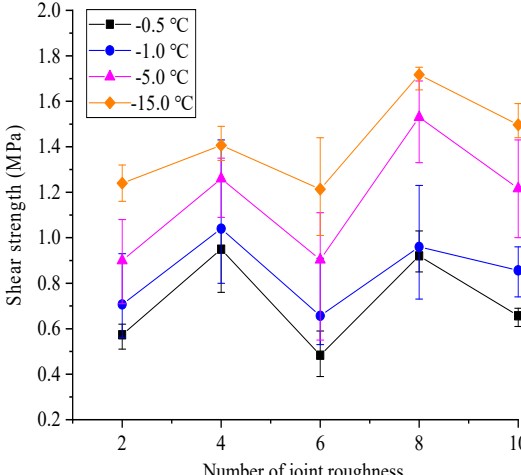


**Figure 4.** Shear strength against joint roughness at different freezing temperatures. Experimental condition: $v$ = 0.2
mm/min, $d$ = 2 mm, $\sigma_n$ = 0.5 MPa.





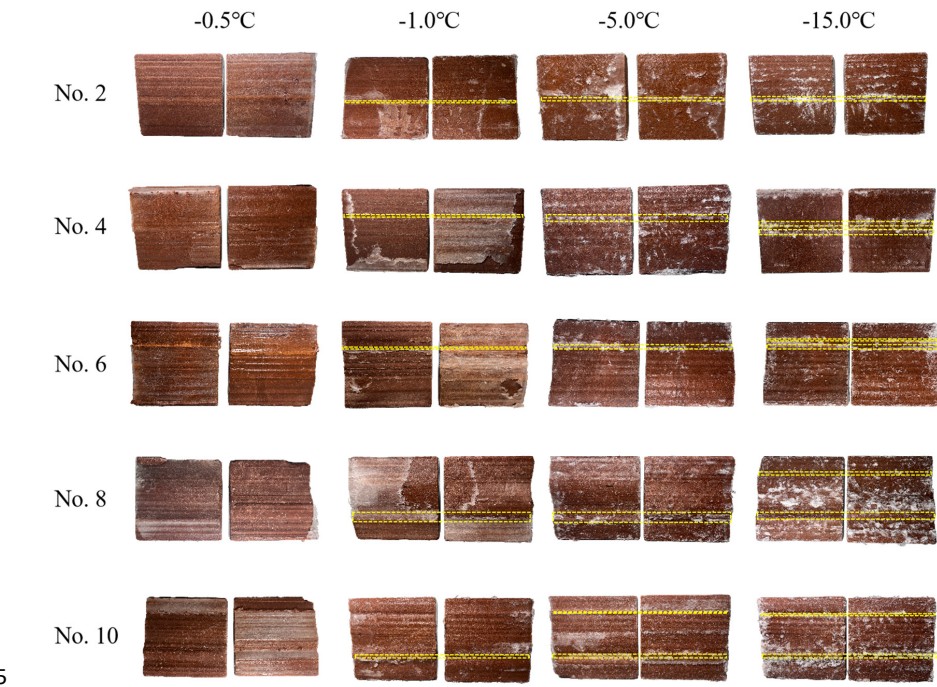

**Figure 5.** Shear rupture modes of ice-filled joints at different freezing temperatures. The yellow lines show the main aggregation of rupture ice. Ice after rupture will aggregate in roughness bulges perpendicular to the shear direction.

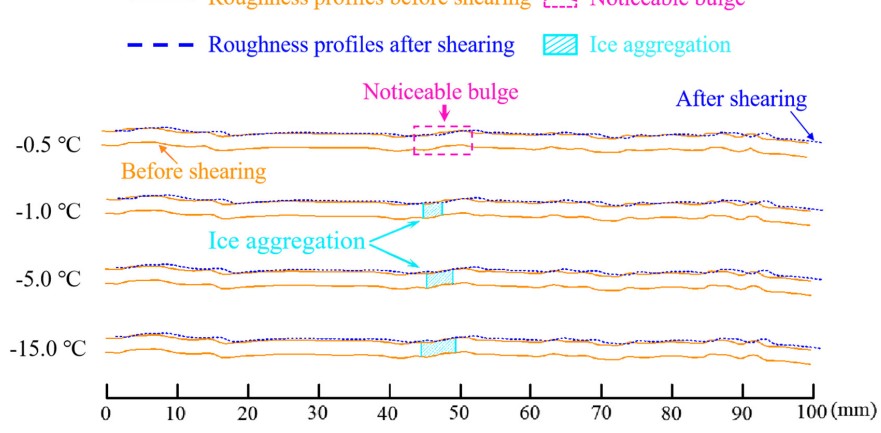



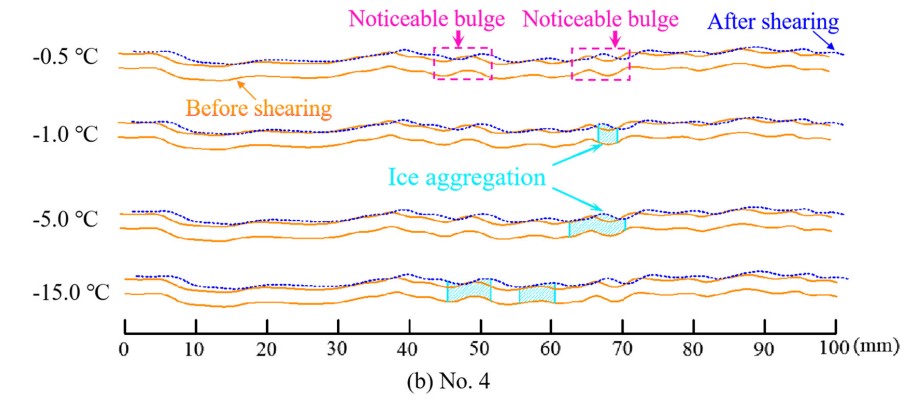


(b) No. 4

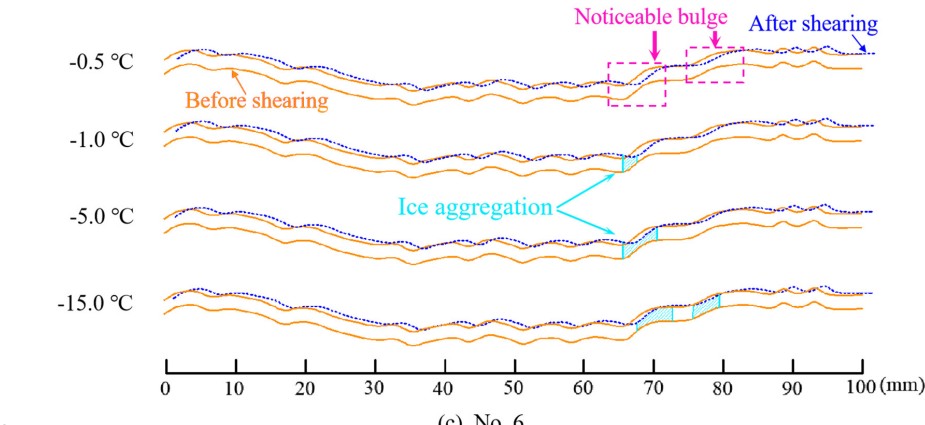


(c) No. 6

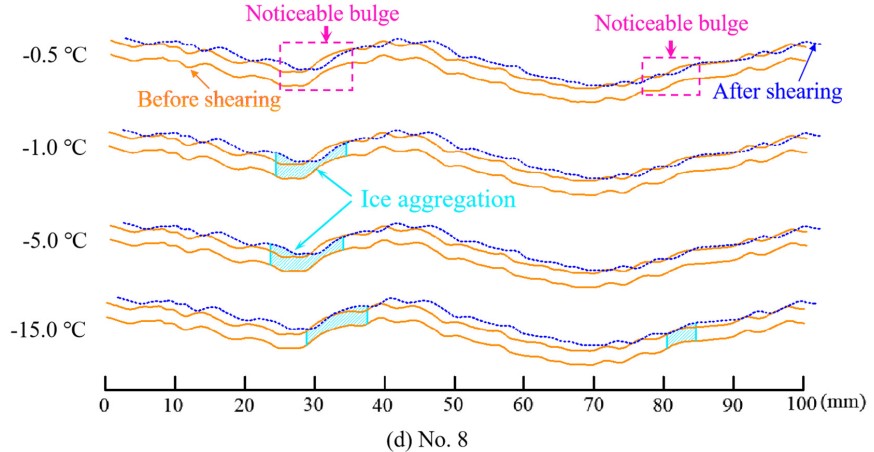


(d) No. 8



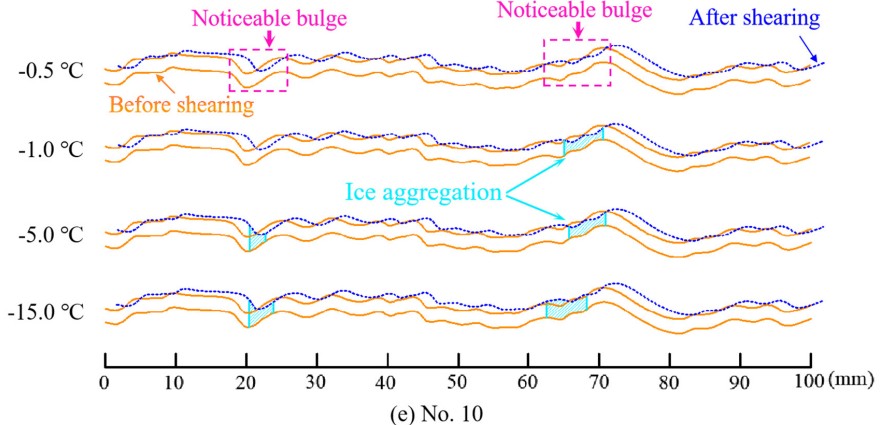

(e) No. 10


**Figure 6.** Shear aggregation areas of ice along the profile of roughness. Experimental condition: $v = 0.2$ mm/min, $d$

= 2 mm, $\sigma_n = 0.5$ MPa. Some blue profiles are located under the orange profiles after shearing, which means the

width of joints becomes smaller. Generally, the reduction of width occurs before some bulges and the rupture ice

will aggregate before these bulges. These bulges are defined as noticeable bulges. Therefore, the bulges causing the

reduction of joint width and aggregation of ice are called noticeable bulges in this study.

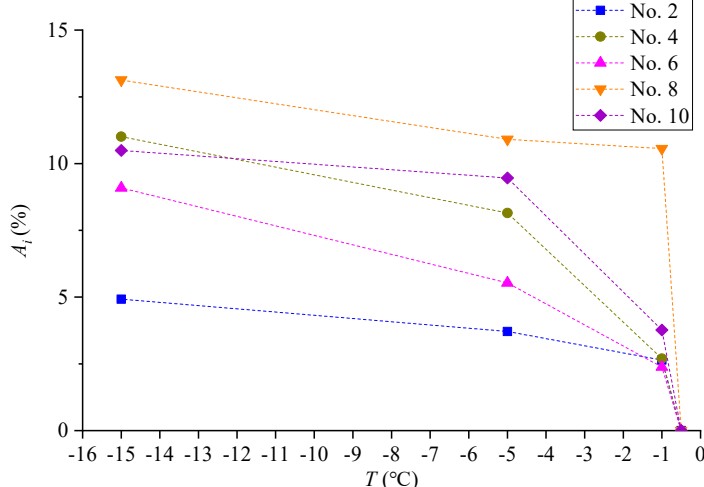





**Figure 7.** Aggregation area of rupture ice increases with the reduction of freezing temperature. Experimental
conditions: $v = 0.2$ mm/min, $d = 2$ mm, $\sigma_n = 0.5$ MPa. $A_i$: aggregation area of rupture ice.
The peak shear displacement and normal displacement also are dependent on the freezing temperature
(Table 3 and Table 4). With the increase of freezing temperature, the peak shear displacement increases
because the joint ice will change from brittle to ductile (Bragov et al., 2015). Ice is brittle at -15 °C and
-5 °C, so the maximum shear displacement before failure is small at this temperature and the shear failure
mode displays brittle characteristics. When the temperature increases to -1 °C, the solid ice becomes to
be ductile, therefore a larger shear displacement arises before failure. However, the shear dilatancy
reduces with increasing the freezing temperature. Solid ice is a kind of temperature-dependent material,
the elastic modulus of which almost linearly decreases with increasing the freezing temperature (Sinha,
1989; Han et al. 2016). The inhibition of normal stress on the shear dilatancy is greater at the high freezing
temperature during shear process.
Several typical shear stress-displacement and normal-shear displacement curves for the profile of No. 4
are plotted in Fig. 8. The ice-filled joint shows significant residual shear strength beyond the peak point,
which slightly decreases with increasing shear displacement. This residual shear strength is caused by
the friction effect between the upper and lower ice-filled blocks. In addition, the normal shear dilatancy
displays increasing trend with shear displacement, which is caused by the climbing effect of ice-filled
joints. It should be noted that the shear strength has a second rising point at the residual strength stage,
because the shear rate is increased from 0.2 mm/min to 1 mm/min in order to accelerate the completion
of the shear process. Schulson and Fortt (2012) claimed that the friction between ice interfaces increases





when the shear rates increase from 0.06 mm/min to 0.6 mm/min. Therefore, the sudden rise of residual
shear strength can be attributed to the accelerated shear rate.

**Table 3.** The peak shear displacement at the peak points of shear strength (mm)

| Profile No. | Freezing temperature | | | |
|---|---|---|---|---|
| | -15 °C | -5 °C | -1 °C | -0.5 °C |
| No. 2 | 1.36 | 1.46 | 1.72 | 1.84 |
| No. 4 | 1.62 | 1.75 | 1.86 | 2.08 |
| No. 6 | 1.33 | 1.53 | 1.71 | 1.83 |
| No. 8 | 1.78 | 1.85 | 1.99 | 2.12 |
| No. 10 | 1.63 | 1.79 | 1.87 | 1.94 |


**Table 4.** The normal shear dilatancy at the point of peak shear strength (mm)

| Profile No. | Freezing temperature | | | |
|---|---|---|---|---|
| | -15 °C | -5 °C | -1 °C | -0.5 °C |
| No. 2 | 0.24 | 0.23 | 0.14 | 0.08 |
| No. 4 | 0.46 | 0.37 | 0.31 | 0.31 |
| No. 6 | 0.27 | 0.28 | 0.22 | 0.12 |
| No. 8 | 0.77 | 0.44 | 0.37 | 0.36 |
| No. 10 | 0.61 | 0.32 | 0.21 | 0.39 |



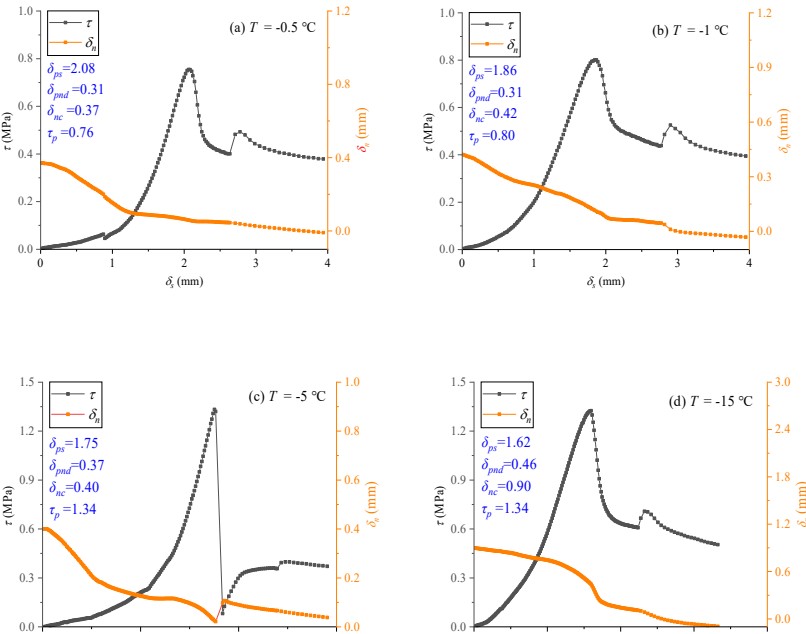



**Figure 8.** Shear strength and normal displacement versus the shear displacement for the profile of No. 4 in the

temperature group. $\delta_{ps}$ and $\delta_{pnd}$ are the shear displacement and normal shear dilatancy at the point of peak shear

strength, $\tau_p$ and $\delta_{nc}$ is the initial compression deformation.

Another finding is that the JRC is not suitable to interpret the influence of joint roughness on the shear

strength of ice-filled joints, because the peak shear strength does not monotonically increase with

increasing JRC index. The peak shear strength displays an increase-decrease-increase-decrease trend

against JRC from No. 2 to No. 10 (Fig. 4). Figure 9 shows that the peak shear strength displays a linear

increasing trend with increasing aggregation areas of fragmented ice after failure. The aggregation area

of fragmented ice can be treated as the effective climbing area which makes a significant contribution to

the improvement of shear strength, because the fragmented ice is produced under compression-shear



stress in the process of climbing the steep bulges. As a consequence, only these steep bulges causing
aggregation of rupture ice contribute to the improvement of shear strength. The variation law of shear
dilatancy against the roughness also is in accordance with the shear strength of ice-filled joints, but it is
different from the change law of JRC (Table 4). In Fig. 6, the gathering of fragmented ice mainly arises
in the front of the steepest bulge. It illustrates that the improvement of shear strength of joint ice is caused
by a part of the steepest bulge instead of the total roughness. Therefore, JCR may be not suitable for the
prediction of shear strength of ice-filled joints. For example, although the JCR of No. 6 is much larger
than No. 4, the effective steep bulge to cause ice aggregation after failure is smaller than that of No. 4
(Fig. 7). This phenomenon confirms that the improvement of shear strength is only caused by some
noticeable steep bulges instead of the total bulges.

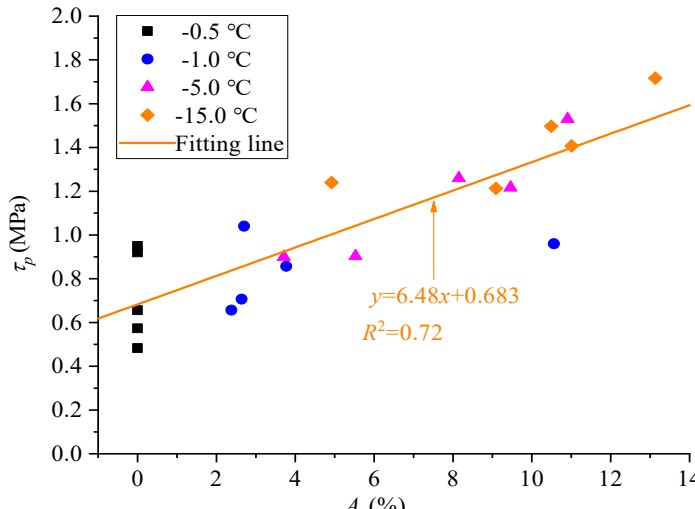


**Figure 9.** Peak shear strength linearly increases with increasing aggregation areas of rupture ice. Experimental
condition: $v$ = 0.2 mm/min, $d$ = 2 mm and $\sigma_n$ = 0.5 MPa.





**3.2 Effect of shear rates**
The shear rates have significant effects on the strength of solid ice as observed in the previous literature
(Petrovic, 2003). Low shear rates are used to conduct quasi-static shear experiments, including 0.2
mm/min, 0.4 mm/min and 0.8 mm/min. Figure 10 shows that the peak shear strength slightly decreases
with increasing shear rates. Solid ice is a kind of typical elasto-plastic material. When the shear rate is
slow, the ice crystal has enough time to shear slip and it will present ductile failure characteristics. At a
low shear rate, the free water on the slip interface will reorganize at the water-ice interface to form ice,
however, it is hard for the ice crystal to adjust to adapt the shear slip at high shear rates, which will cause
the shear rupture of ice crystals and hinder the growth of ice on the water-ice interface (Lou et al., 2019).
Figure 11 shows that a high shear rate will induce brittle failure of joint ice and more fragmented ice
crystals are produced. As a result, the shear strength reduces with increasing shear rates from 0.2 mm/min
to 0.8mm/min. The previous literature shows that there is a critical loading rate for the transition from
ductile to brittle behavior of polycrystalline ice (Timco and Frederking, 1982; Gold, 2018). In this study,
the transition point of ice-filled joint is not definitely derived due to the limitation of the shear rate range.



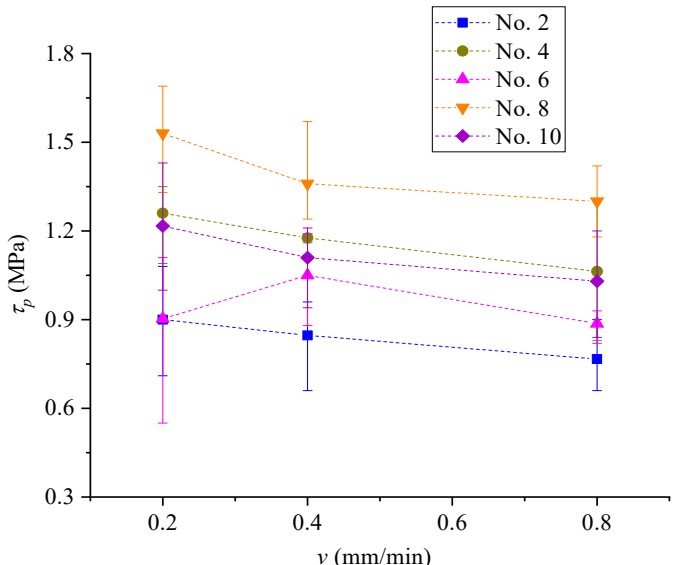


**Figure 10.** Effect of shear rate on the peak shear strength. Experimental condition: $T = -5$ °C, $d = 2$ mm and $\sigma_n = 0.5$

MPa.



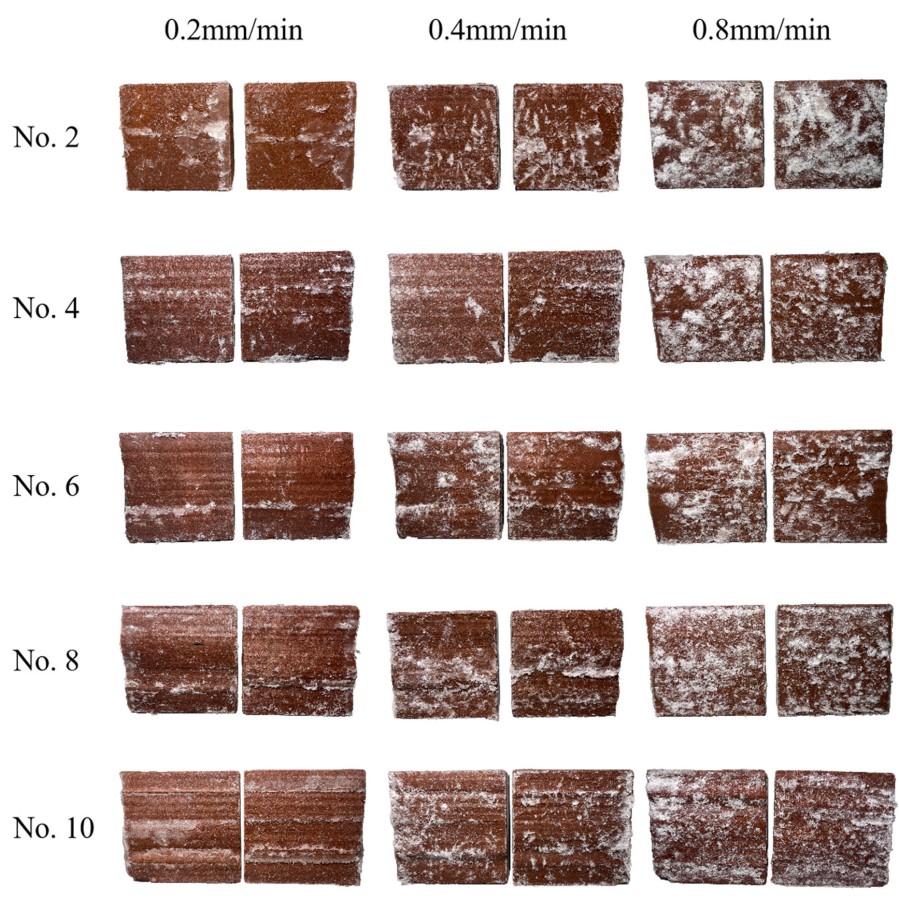

**Figure 11.** The shear rupture characteristics of joint ice under different shear rates. Experimental condition: $T = -5$ ℃, $d = 2$ mm and $\sigma_n = 0.5$ MPa. The ice crystal that cannot adapt to shear slip at high shear rates will form brittle failure. The joint ice of brittle failure shows more micro fractures which make it more reflective. This will cause a white appearance of the rupture ice on the joint surface. The ductile failure of ice displays a transparent appearance without white color, which is hard to observe. Therefore, a larger area of white appearance implies a much more serious brittle failure of joint ice.



### 3.4 Effect of joint openings


Joint opening is another critical factor influencing the shear strength of ice-filled joints. The maximum
height difference of the standard JRC curves suggested by Barton and Choubey (1977) is approximately
2.14 mm, 2.40 mm, 6.24 mm, 6.85 mm and 4.48 mm for the profiles of No. 2, No. 4, No. 6, No. 8 and
No. 10, respectively. The joint openings are chosen as 2 mm, 8 mm and 14 mm because 2 mm is smaller
than all the maximum height differences while 14 mm is much larger than them. The rupture
characteristics of joint ice against the joint opening are plotted in Fig. 12. When the joint opening is 2
mm, the aggregation phenomenon of rupture ice is evident. However, the aggregation phenomenon
disappears for the profiles of No. 2, No. 4 and No. 6 when the joint opening is 8 mm. When the joint
opening increases to 14 mm, there is not any aggregation of rupture ice arising for all the joints. Figure
13 shows that when the joint opening increases from 2 mm to 14 mm, the shear strength of ice-filled
joints decreases. The shear strength of pure solid ice also is measured in the laboratory, which is
approximately 0.83 MPa on the condition that $T = -5$ °C, $v = 0.2$ mm/min and $\sigma_n = 0.5$ MPa. When the
joint opening is 14 mm, the shear strengths of ice-filled joint are approximately 0.83 MPa and they are
independent of the joint roughness. When the joint opening is 8 mm, the shear strengths of ice-filled joint
are very close to the shear strength of pure solid ice (0.83 MPa) for the joint of No. 2, No. 4 and No. 6.
The reason is that 8 mm has exceeded the critical filling thickness of these joints (No. 2, No. 4 and No.
6), therefore the shear strength of these ice-filled joints is only controlled by the solid ice instead of joint
roughness. In addition, there is not any significant ice aggregation on the joint surfaces of No. 2, No. 4
and No. 6 when the joint opening is 8 mm, and the shear failure happens inside the joint ice. However,
for the ice-filled joints of No. 8 and No. 10, the shear strengths are larger than 0.83 MPa, which illustrates



that the critical filling thickness for the profiles of No. 8 and No. 10 should be larger than 8 mm but
smaller than 14 mm. There is aggregation ice arising before large bulges, and these large bulges would
prevent the direct shear failure of joint ice and improve the shear strength.
The influence of joint opening and roughness on the shear strength can be explained by using the shear
failure path of ice-filled joints as shown in Fig. 14. When $d$=2 mm, the shear climbing will occur before
some large bulges for all the joint profiles. This climbing action induces the aggregation of rupture ice
and change of shear path. As a consequence, the shear strength will improve. When $d$=8 mm, the shear
failure path will not be disturbed for the profiles of No. 2, No. 4 and No. 6, however, the shear failure
path changes due to the climbing action for the profiles of No. 8 and No. 10, in which a significant
aggregation of rupture ice is produced. Therefore, the shear strengths of ice-filled joints for the profiles
of No. 2, No. 4 and No. 6 are approximately equal to the solid ice, while the shear strengths for the
profiles of No. 8 and No. 10 are much larger than 0.83 MPa. When $d = 14$ mm, the shear failure happens
inside the joint ice for all joint profiles, therefore, the shear failure path and shear strength will not be
influenced by the joint roughness and no aggregation of rupture ice occurs. The shear dilatancy
deformation of the ice-filled joints in Fig. 15 has further proved the climbing actions, including all the
profiles with joint opening of 2 mm, and the profiles of No. 8 and No. 10 with joint opening of 8 mm.
The climbing effect of the No. 2 ice-filled joint with opening of 2 mm is not remarkable, therefore the
shear dilatancy is very small and the shear strength also is close to pure solid ice (0.83 MPa). Regardless
of the critical filling thickness, the present study shows that the shear strength of ice-filled joints
decreases with increasing joint openings from 2 mm to 14 mm, and it is related to the joint roughness
below the critical infilling thickness. When the filling ice exceeds the critical thickness, the shear strength
of ice-filled joints is equal to the shear strength of solid ice under the same condition. It should be noted
that the critical filling thickness for each roughness will be determined in future studies.

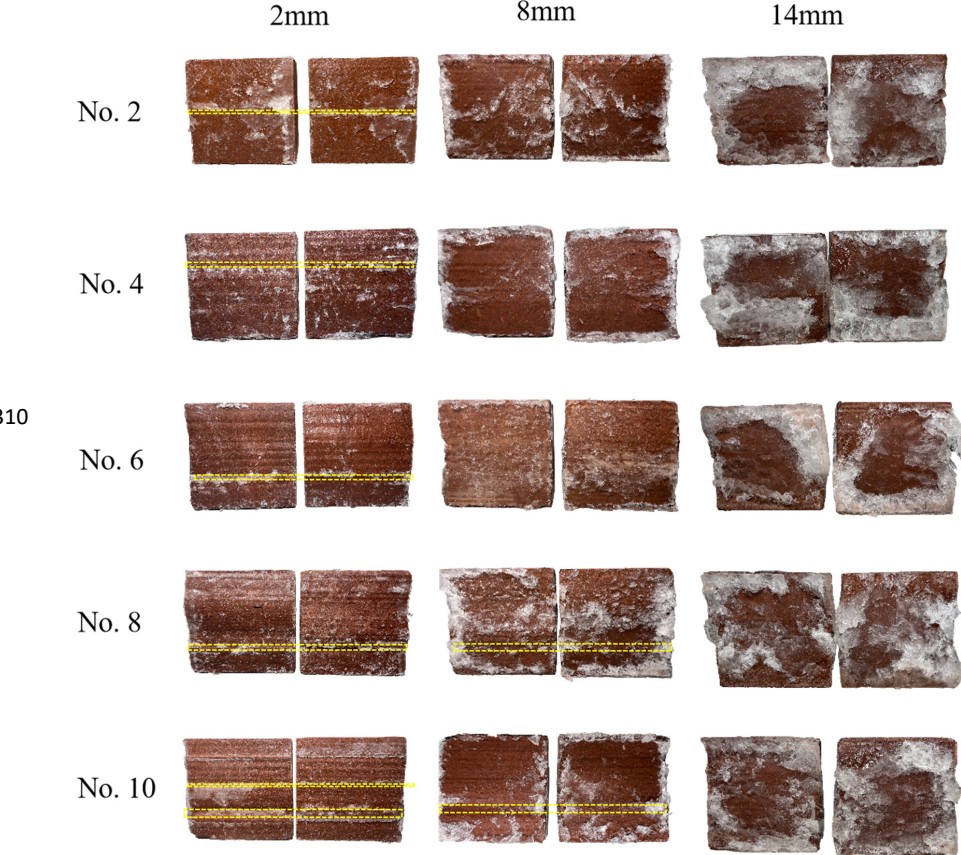

**Figure 12.** The shear rupture characteristics of ice-filled joints with different openings. Experimental condition: $T =$
-5 ℃, $d = 2$ mm and $\sigma_n = 0.5$ MPa. The yellow lines show the main aggregation of rupture ice. Ice after rupture will
aggregate in roughness bulges perpendicular to the shear direction. The aggregation phenomenon disappears as the
joint openings increase. The aggregation phenomenon of profiles No. 2, No. 4 and No. 6 disappear in 8 mm joint
openings. All profiles' aggregation phenomena disappear in 14 mm joint openings.



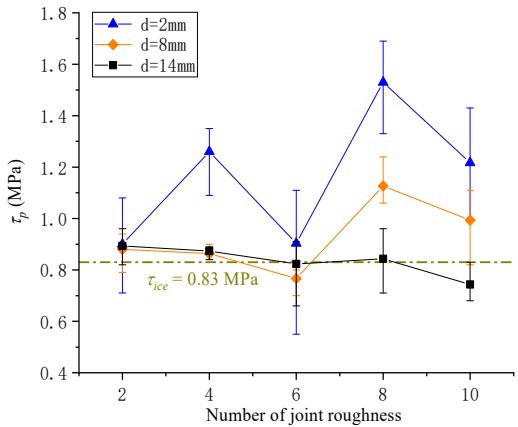


**Figure 13.** Effect of joint opening on the peak shear strength. Experimental condition: $T$ = -5 °C, $v$ = 0.2 mm/min


and $\sigma_n$ = 0.5 MPa.



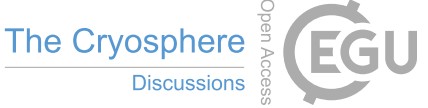

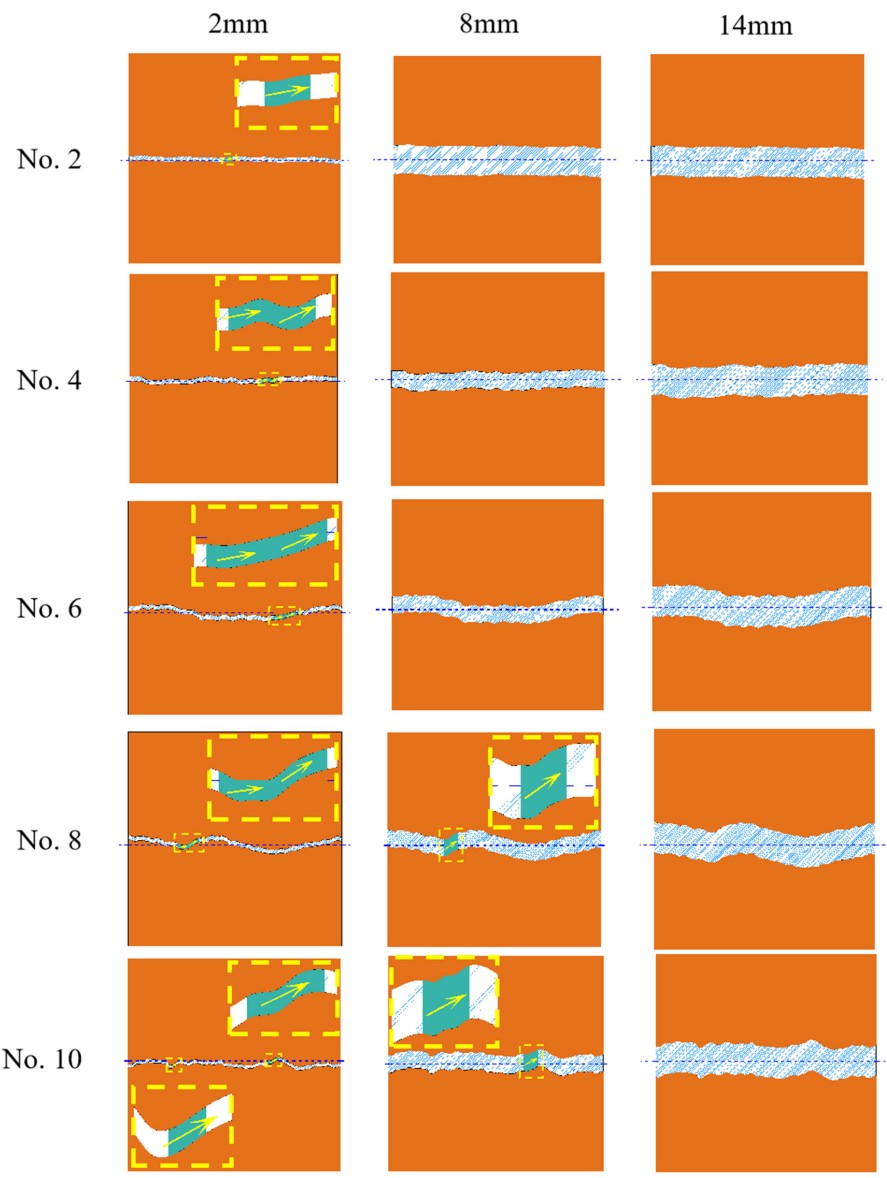


**Figure 14.** Influence of joint roughness on the shearing slip path. Experimental condition: $T$ = -5 ℃, $v$ = 0.2 mm/min

and $\sigma_n$ = 0.5 MPa.

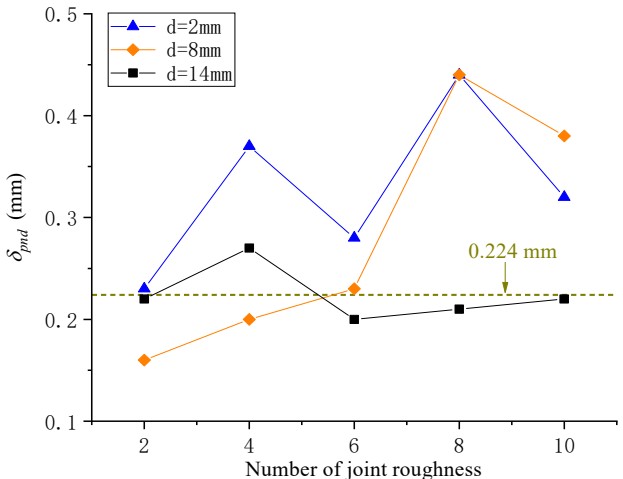

**Figure 15.** Effect of joint opening on the shearing dilatancy. Experimental condition: $T$ = -5 °C, $v$ = 0.2 mm/min and

$\sigma_n$ = 0.5 MPa.

### 3.5 Effect of normal stress

The normal stress group was used to investigate the effect of normal stress on the shear strength of ice-

filled joints, including 0 MPa, 0.5 MPa, 1.0 MPa, 1.5 MPa and 2.0 MPa. The shear strength of ice-filled

joints displays a significant increasing trend with increasing normal stress (Fig. 16). The Mohr-coulomb

criterion may be used to express the relationship between the shear strength and normal stress as below:

$$\tau_p = c_j + \sigma_n \tan \phi_j \tag{1}$$

Where $\tau_p$ = shear stress on plane, $\sigma_n$ = normal stress on plane, $c_j$ = cohesion of ice-filled joints,

$\phi_j$ = internal friction angle of ice-filled joints.

Figure 16 shows Mohr-coulomb criterion can be well used to calculate the shear strength of ice-filled

joints against the normal stress. The shear rupture modes of the joint ice are given in Fig. 17. A

remarkable ice aggregation phenomenon can be found on the surface of joints and the aggregation occurs





at a stable location of the joint profile regardless of the normal stress. The aggregation area of rupture ice
increases with increasing normal stress, because climbing bulges is harder and the solid ice is easier to
be crush at the front of large bulges under the higher normal stress (Fig. 18). In Section 3.1, it has
illustrated that the aggregation area of rupture ice is an important index to reflect the shear strength of
ice-filled joints at different freezing temperatures. Actually, the shear strength also linearly increases
with increasing the aggregation area of rupture ice under different normal stress as shown in Fig. 19. It
further illustrates that only some large bulges causing the aggregation of rupture ice can contribute to the
improvement of shear strength instead of the total roughness index, such as JRC.

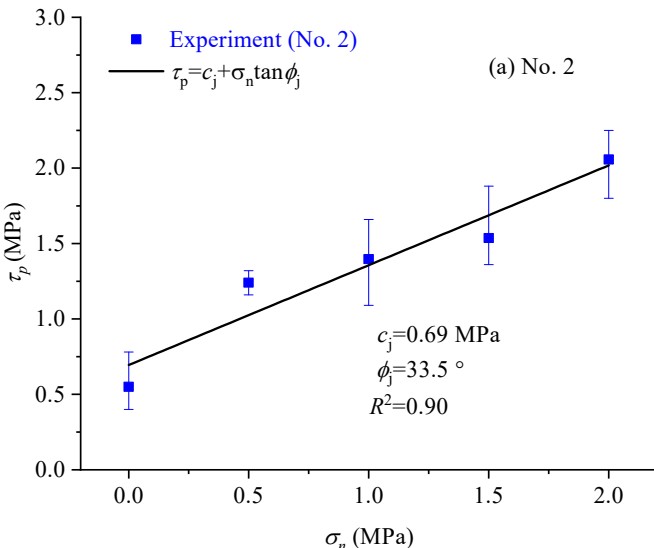






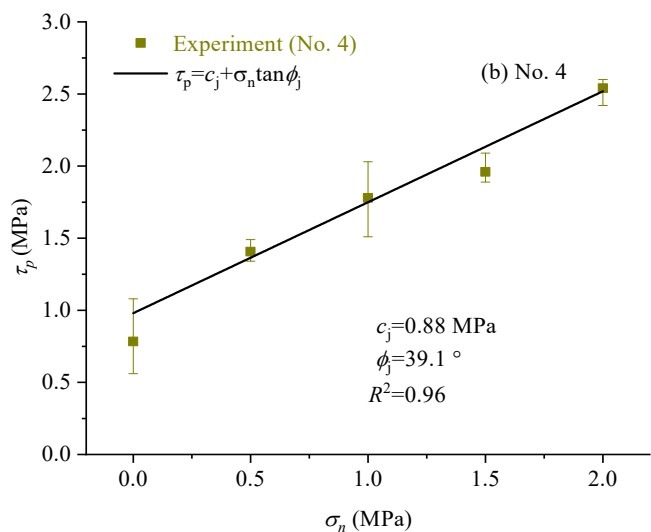


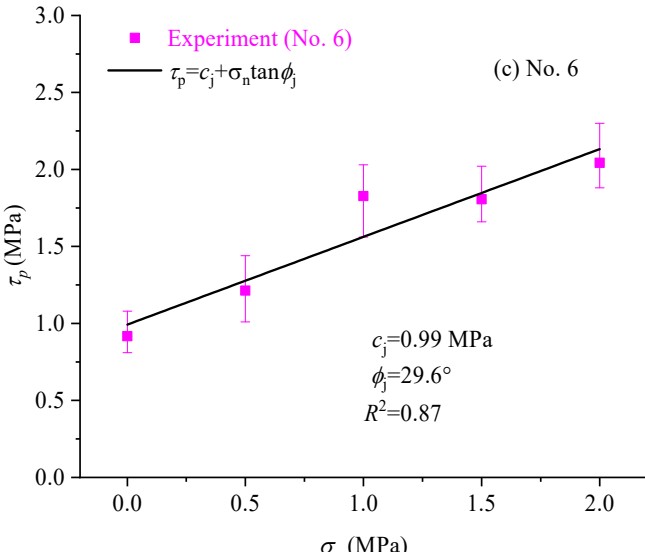






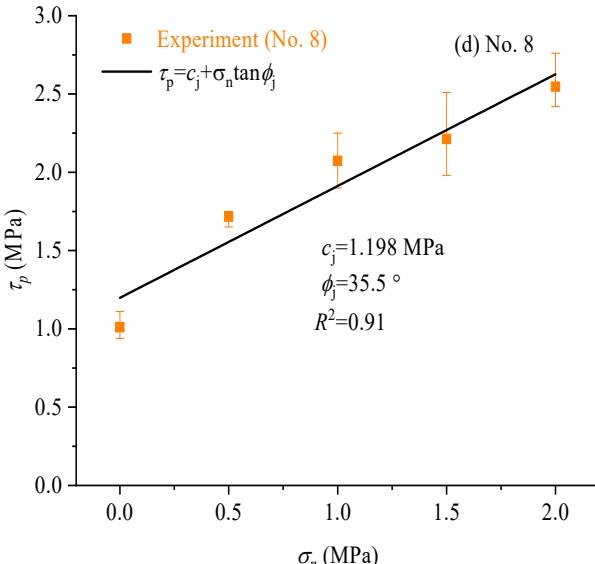


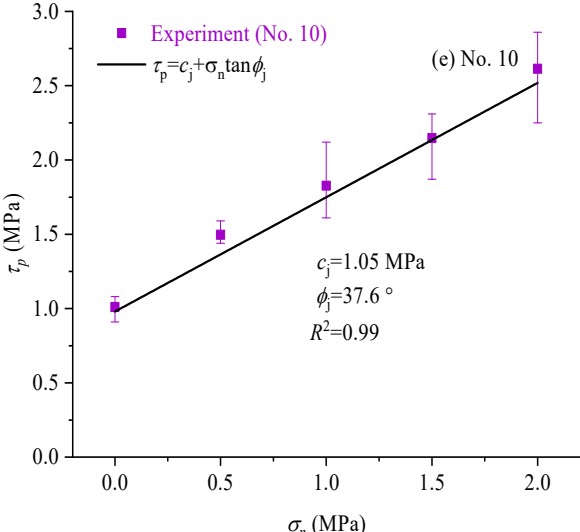


**Figure 16.** Effect of normal stress on the peak shear strength of ice-filled joints. Experimental condition: $T$ = -15 ℃,
$v$ = 0.2 mm/min and $\sigma_n$ = 0.5 MPa.




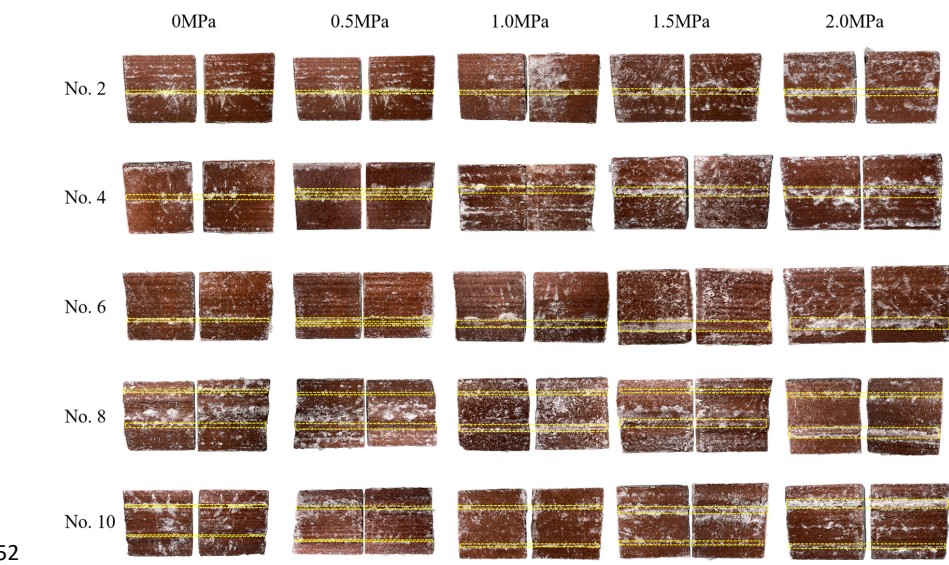

**Figure 17.** Aggregation of rupture ice under different normal stresses. Experimental condition: $T$ = -15 ℃, $d$ = 2

mm and $v$ = 0.2 mm/min. The yellow lines show the main aggregation of rupture ice.

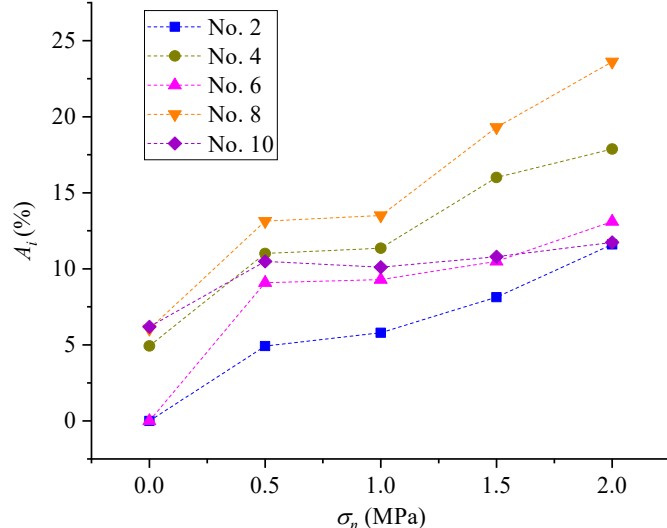

**Figure 18.** Aggregation area of rupture ice increases with increasing normal stress. Experimental condition: $T$ = -

15 ℃, $d$ = 2 mm and $v$ = 0.2 mm/min.



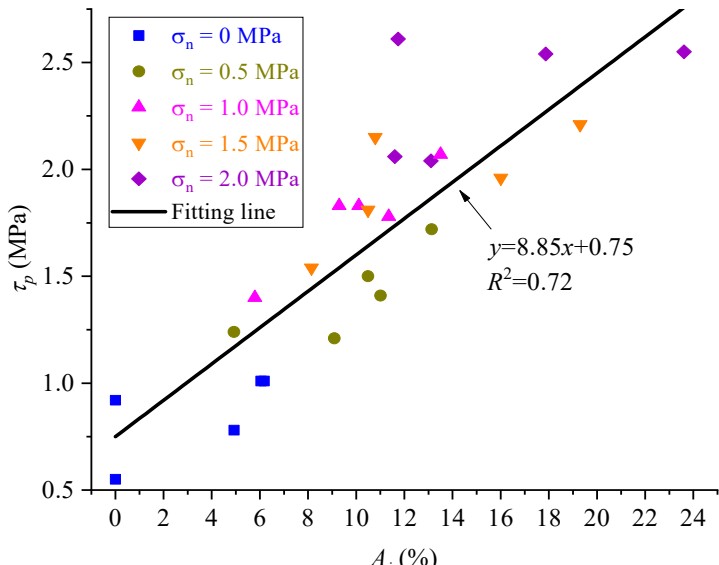

**Figure 19.** Peak shear strength linearly increases with increasing aggregation areas of rupture ice. Experimental

condition: $T$ = -15 ℃, $d$ = 2 mm and $v$ = 0.2 mm/min.

**4. Discussion**

**4.1 The warming degradation mechanism of ice-filled joints**

In this paper, the influence of freezing temperature, shear rate, joint opening and normal stress on the

shear strength of ice-filled joints in rock masses was comprehensively investigated by experiments. The

shear strength remarkably reduces with increasing freezing temperature, because the shear strengths of

solid ice and ice-rock interface decrease with increasing temperature. In order to deeply understand the

warming degradation mechanism of ice-filled joints, the shear strength of pure ice and ice-rock bonding

interface under different freezing temperatures also were tested in this study (Fig. 20).


The test results show that the shear strength of smooth ice-rock bonding interface is larger than that of
pure solid ice at the freezing temperature from -15 to -0.5 °C (Fig. 20a). It implies that the shear failure
should be inside the solid ice instead of ice-rock interface. When the freezing temperature increase from
-1 °C to -0.5 °C, the shear strengths of the ice-rock interface and the solid ice reduce very quickly. Jia et
al. (2015) also claimed the same change law of solid ice against the temperature.
However, the experimental results show that the shearing failure of many rough ice-filled joints at -0.5 °C
is the debonding of ice-rock interfaces (Figs. 5, 11, 12, 17). More shear experiments were carried out on
rough ice-rock interfaces with profiles of No. 4 and No. 8 on the same experimental condition ($\sigma_n$ = 0.5
MPa, $v$ = 0.2 mm/min). It shows that the shear strength of rock-ice-rock "sandwich" is a little larger than
that of ice-rock interface, although the change laws of them against temperature are very similar. Another
novel finding is that the shear strength of ice-rock interface is larger than the shear strength of solid ice
itself below -1 °C (Fig. 20b). Therefore, the shear failure below -1 °C displays the cracking of joint ice
instead of ice-rock interface, and some aggregation areas of rupture ice occur before large bulges (Figs.
5, 11, 12, 17). However, the shear strength of solid ice is larger than that of ice-rock interface above -
1 °C. This is the main reason for the shear failure of rough ice-filled joints along ice-rock interfaces at -
0.5 °C. The freezing temperature of -1 °C is the transition point of shear failure modes. Figure 21 presents
that the shear failure is along the ice-rock interface when the freezing temperature is approximate -0.5 °C,
however, the area of ice attached to the joints has a great increment with the decrement of freezing
temperature from -0.5 °C to -15 °C. It further illustrates that the shear strength of rough ice-rock interface
is larger than that of the solid ice below -5°C. Mamot et al. (2018) also found that the shear failure modes
of the smooth ice-filled joints changed from shearing cracking of joint ice to the debonding of ice-rock





interface when the freezing temperatures increased from -10 ℃ to -0.5 ℃. The smooth joints have a little
ability to resist the shear slide of ice-filled joints. Mamot et al. (2018) claimed that three shear failure
modes may arise between -5 ℃ to -1 ℃, including the debonding of ice-rock interface, shear cracking
of joint ice and their mixed mode. However, only the shear cracking of joint ice occurs at -5 ℃ to -1 ℃
in this study. Therefore, the joint roughness has an effect on the shear strength of ice-filled joints and the
shear failure modes.

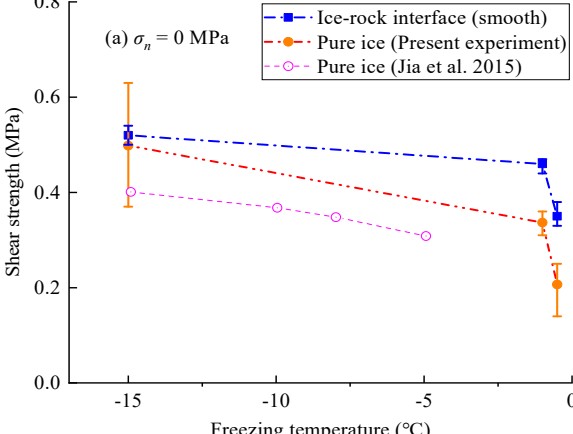





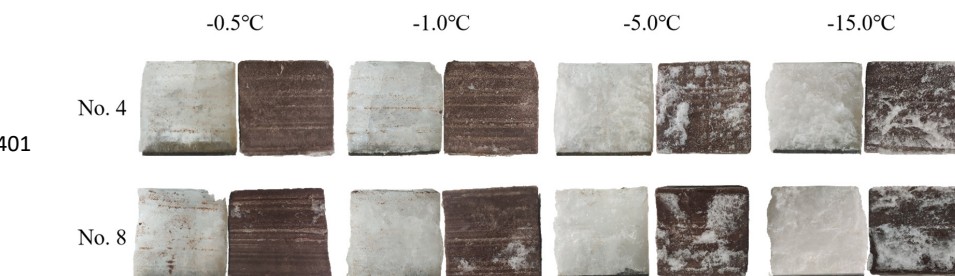

**Figure 20.** Influence of freezing temperature on the direct shear strength of ice and ice-filled joints. Experimental
condition: $v$ = 0.2 mm/min.

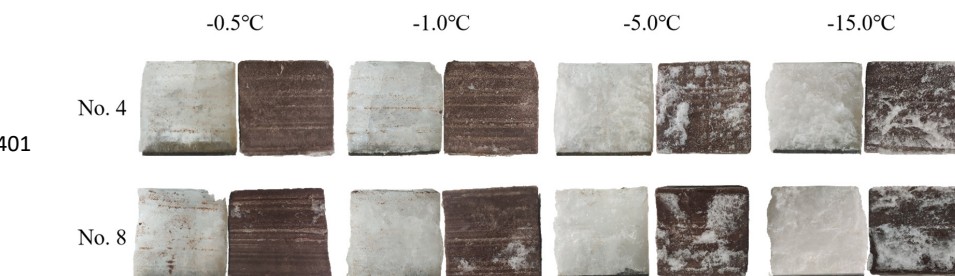


**Figure 21.** Shear failure characteristics of ice-rock interfaces under different temperatures. Experimental condition:
$v$ = 0.2 mm/min, $\sigma_n$ = 0.5 MPa.

**4.2 The coupled effect of joint roughness, opening and normal stress**
The shear strength of smooth ice-filled joints were investigated by Mamot et al. (2018). They found that
the shear strength of smooth ice-filled joints also linearly increases with decreasing temperatures.



Actually, the roughness is another important factor influencing the shear strength of ice-filled joints,
which can improve the ability to resist the shear slide of joints (Fig. 22). The shear strength of the No. 2
ice-filled joint is much smaller than that of No. 8 and No. 10 joints. For the profile of No. 2, the shear
strength of ice-filled joint is approximately equal to that of the solid ice when the normal stress is less
than 1.5 MPa, because the joint opening of 2 mm also is very close to the maximum height difference.
Therefore, the joint opening will determine the effect of joint roughness. However, the shear strength of
solid ice is much smaller compared with the shear strength of ice-filled joints when the normal stress is
2 MPa. It is observed that this normal stress has caused some vertical micro-cracks inside the solid ice.
For the ice-filled joints, the compression damage maybe not remarkable, because both the adhesion of
ice-rock interface and bulges will prevent the lateral expansion of solid ice under high normal stress. A
larger roughness may provide a much stronger confining effect on the lateral expansion. Although the
shear strength increases with increasing JRC number in general, the quantitative relationship between
them are hard to determine. Figure 4 shows that the change of shear strength against the JRC number is
fluctuating. A novel finding of this study is that the aggregation area of rupture ice before large bulges
can be well used to predict the shear strength of ice-filled joints. However, it should be noted that a new
index of roughness should be proposed in future research in order to build the shear strength model
considering joint roughness.
In addition, if the joint opening exceeds the critical value, the influence of joint roughness on the shear
strength of ice-filled joints will disappear. For example, when the thickness of joint ice exceeds 14 mm,
the shear strength of all the ice-filled joints is equal to the shear strength of infilling ice. Section 3.4 has



illustrated that the value of critical joint opening is depended on the maximum height different of the
joint, which need to study further.

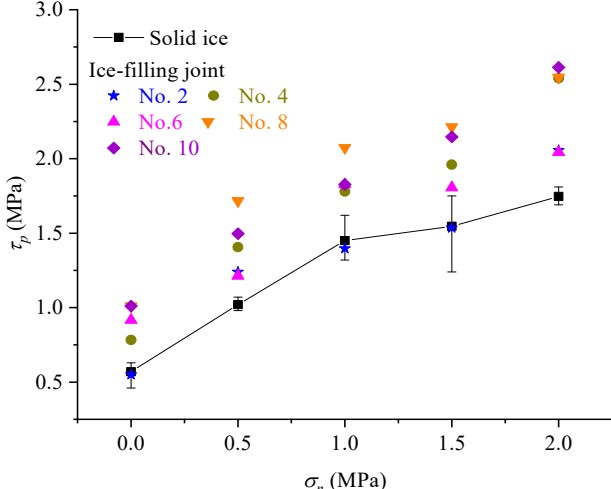


**Figure 22.** Shear failure characteristics of ice-rock interfaces under different normal stress. Experimental condition:
$v$ = 0.2mm/min, $d$ = 2 mm, $T$ = -15 ℃.

**4.3 Potential application for prediction of rock avalanches in a warming climate**
In recent years, there are many large rock avalanches occurred in the Alps. The rock avalanches that
occurred on the Brenva galcier, the Punta Thurwieser and the Drus are some of the recent examples,
which have strong impacts on the high mountain infrastructure stability and landscape evolution (Mamot
et al., 2018). The rock avalanches are related to the degradation of bedrock permafrost and ice-filled
joints. Our study shows that the peak shear strength of ice-filled joints increases with the increase of
roughness and normal pressure. This implies that the rockfall will be more stable with higher roughness
and normal pressure. In addition, when the joint openings increase, the peak shear strength will decrease,



and large joint openings will reduce the effect of joint roughness. The peak shear strength of ice-filled
joints decreases with the increase of freezing temperature. Moreover, when the freezing temperature is
close to 0 °C, the pre-melting of ice-rock interface induced by the normal stress will cause a reduction of
bonding strength. This result can explain the phenomenon that the boundary of ice-filled joint between
frozen and unfrozen become unstable, especially in summer. The peak shear strength of ice-filled joints
decreases with the increase of shear rate. It is hard for the ice crystal to adjust to adapt the shear slip at
high shear rates so the rockfall may happen.
As the global temperature rises, collapse disasters of ice-filled rock mass caused by warming and thawing
often occur in permafrost regions. A constitutive model can be further constructed according to the
experiment results. Then combining with a numerical software, this constitutive model can be used to
predict the disaster of rock avalanches in the cold region in the future research. Although Mamot et al.
(2018) has established a constitutive model for joints, the constitutive model only considers temperature
and normal stress, however, the influence of the joint roughness, opening and shear rate is ignored.
Through our study, it is evidenced that the joint roughness, shear rate, joint opening and temperature are
physical quantities that must be considered in the constitutive model. A constitutive model including
these physical quantities will be proposed in our future research.
**5 Conclusions**
Above all, this study has provided a comprehensively experimental study on the shear process of ice-
filled joints, considering the influence of freezing temperature, joint roughness, shear rate, joint opening
and normal stress. The following conclusions can be drawn based on this research:



(1) The shear strength of ice-filled joints decreases with increasing temperature. The shear failure mode
change from shear cracking of joint ice to the debonding of ice-rock interface when the temperature
increases to -0.5 ℃, because the bonding strength of ice-rock interface is less than that of solid ice at -
0.5 ℃ ($v = 0.2$mm/min, $\sigma_n = 0.5$ MPa).
(2) The joint roughness can improve the shear strength of ice-filled joints. The shear strength of ice-filled
joints linearly increases with increasing the aggregation area of rupture ice before some large bulges.
However, the relationship between the JRC index and the shear strength is poor. In addition, the effect
of joint roughness is related to the joint opening and normal stress.
(3) The shear strength of ice-filled joints decreases with increasing joint opening. When the joint opening
increases from 2 mm to 14 mm, the aggregation of rupture gradually disappears and the shear strength
of ice-filled joint is equal to that of solid ice. Therefore, the joint roughness does not make any
contribution to the shear strength when the joint opening exceeds a critical value, which is related to the
maximum height difference of joint surface.
(4) The shear strength of ice-filled joints decreases when the shear rate increase from 0.2 mm/min to 0.8
mm/min. The infilling ice will change from ductile failure to brittle failure by observing the rupture ice
on the joint surface. The aggregation area of rupture ice also decreases while the brittle rupture
phenomenon is more serious with increasing shear rate.
(5) The shear strength of ice-filled joints linearly increases with increasing normal stress, which well
satisfies the Mohr-coulomb criterion. The aggregation area of rupture ice also increases with increasing
normal stress. In addition, the improvement of shear strength of the ice-filled joints caused by normal



stress is much larger that of solid ice, because the bulges of the joint surface can prevent the lateral
expansion of ice under compression.
**Acknowledgements**
This work was supported by National Natural Science Foundation of China (Grant No. 42072300 and
No. 41702291), Project of Natural Science Foundation of Hubei Province (Grant No. 2021CFA094).
**Conflict of interest**
The authors declared that they have no conflicts of interest to this work.

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
