# Peer review of "The temperature-dependent shear strength of ice-filled"

_The Cryosphere, 2022_

## Author Comment (AC2)

Nov 20, 2022

Dear referees

Thanks for your comments and suggestions. Based on these comments and suggestions, we have made careful modifications on this manuscript. Appended to this letter is our point-to-point response to the comments. These comments were reproduced and our response were given directly afterward in a different color (blue).

We hope that the manuscript can be accepted for publication in **The Cryosphere**. If you have any questions, please contact us immediately. We are grateful for your attention to our manuscript. Once again, thanks very much for your arduous work and instructive suggestions to our manuscript processing.

Sincerely,

Dr. Shibing Huang
* * *
School of Resources and Environmental Engineering,

Wuhan University of Science and Technology,

Wuhan, 430081, P.R. China

Telephone: +86 185-0275-5916

E-mail: huangshibing@wust.edu.cn

Response to referee:

1. Very few results have been included in the abstract part. Move L14-15 to the introduction part. The abstract should be concise and include the main results of this study.

Response: Thank you for your good suggestion. We have deleted L14-15 in the original paper, because it does not belong to this part. The introduction part also has similar content about the important factors, therefore, it is not added again (Lines 71~72 in our revised paper). In addition, more important results have been added in the abstract, mainly including the effect of joint roughness, freezing temperature, joint opening and normal stress. (Details can be found in the abstract in our revised paper, which are marked in red.)

2. L19-20, "when the joint opening is large enough", what is the threshold in terms of joint opening in this study?

Response: For the joints, different rough profiles have different joint opening thresholds. Generally, a rougher joint has a larger critical joint opening, beyond which the effect of the roughness on the shear strength of ice-filled joints will be not significant. In this study, the thresholds of joint openings for the No. 2 and No. 6 profiles are between 2-8 mm, and the thresholds for No. 4, No. 8 and No. 10 are between 8-14 mm. If we want to determine the exact thresholds of joint opening for these joint

profiles, more experiments should be conducted on the ice-filled joints by refining the joint openings, which will be investigated in the future research. In order to avoid misunderstanding, this sentence has been changed to "As the joint opening increases, the effect of joint roughness decreases and the shear strength of ice-filled joints tend to be equal to the shear strength of pure ice." (Details can be found on Lines 18~19 in our revised paper, which are marked in red.)

3. L84, the meaning of the symbols can be included in the table

Response: Thank you for your good suggestion. We have added the meaning of the symbols in Table. 1 and deleted the description in the table title. (Details can be found in Table. 1 in our revised paper, which are marked in red.)

4. In preparing the samples, the line roughness profile is simply extended to a surface roughness profile. Therefore, the authors should clarify the shearing direction in terms of the profile.

Response: Thank you for your good suggestion. We have added a new figure to clarify the shearing direction. All the samples are sheared along the directions in Figure 2. (Details can be found in Figure 2 in our revised paper, which are marked in red.)

[Figure]

Figure 2. The shear directions for different joint profiles.

5. L150, it is suggested to provide the corresponding references.

Response: Thank you for your good suggestion. We have provided the corresponding references on Lines 154 to 155. The shear strength of soil-filled and cement-filled joints increases with increasing JRC index, which has been presented in the references (Xu et al. 2012; Zhao et al. 2020). (Details can be found on Lines 155~157 in our revised paper, which are marked in red.)

Xu, D. P., Feng, X. T., Cui, Y. J.: A simple shear strength model for interlayer shear weakness zone. Eng. Geol., 147, 114-123, http://dx.doi.org/10.1016/j.enggeo.2012.07.016, 2012.

Zhao, Y. L., Zhang, L. Y., Asce, F., Wang, W. J., Liu, Q., Tang, L. M. and Cheng, G. M.: Experimental Study on Shear Behavior and a Revised Shear Strength Model for Infilled Rock Joints, Int. J. Geomech., 20(9), 04020141, 10.1061/(ASCE)GM.1943-5622.0001781, 2020.

6. L151, avoid using "law" for describing the changes. In addition, "as shown in Fig. 5" should be "show in Fig. 5" or ", as shown in Fig. 5". It is suggested to double-check the English carefully.

Response: Thank you for your good suggestion. We have done a double check about the English and fixed grammatical errors. "law" is replaced by "trend". "as shown in Fig. 6" has been corrected as ", as shown in Fig. 6". (Details can be found on Lines 157 to 158 in our revised paper, which are marked in red.)

7. In Fig. 5, present the shear direction in the photos. In addition, the aggregation of rupture ice is not clearly seen from these photos. How is it determined?

Response: Thank you for your good suggestion. We have presented the shear direction in Figure 5, which is Figure 6 in our revised paper. The aggregation of rupture ice is defined by two main conditions: ① the rupture ice have a white appearance with obvious rupture characteristics by enlarging the pictures; ② the rupture ice appeared before the specified noticeable bulges, which can be determined easily. As shown in the following figure, we can draw the location of the aggregation ice in the standard roughness curves by comparing with the shear failure picture of the ice-filled joint. Then the aggregation area of the rupture can be estimated because the joints are two-dimensional surfaces. It should be noted that this is a simple and approximate estimation method. Due to the

limitation of the length of this text, the picture presented in Fig. 6 is a little small, therefore, we have drawn the aggregation area with green lines. This point has been explained in our revised paper. On the left of the green box in Fig. 6b, some rupture ice is not included in the aggregation area, because these rupture ice is caused by the extrusion of the original joint ice from the box. (Details can be found on Line 161~170 and in Fig. 6b in our revised paper, which are marked in red.)

[Figure]

Ice is extruded from the green box under shearing process

Figure 6b Determination of the aggregation area of the rupture ice

8. In Fig. 6, please clarify the definition or the standards for "noticeable bulge" in detail.

Response: The "noticeable bulges" are pointed out in Figure 7 in our revised paper. The bulges causing the reduction of joint width and aggregation of ice are called noticeable bulges. The noticeable bulges

have larger inclination angles and they are far away from the joint edges. The definition of the noticeable bulge is added in our revised paper (Details can be found on Lines 206 to 208 in our revised paper, which are marked in red.)

9. In Fig. 7, how is the aggregation of rupture ice experimentally determined? Please provide methods and details. In addition, is the data point a mean value of the three samples?

Response: When the shear experiments are completed, we will take a photo of the joint surface by using a camera. The rupture ice has a white appearance, low transparency and obvious rupture characteristics by observing the enlarged pictures of the ice-filled joints after shear failure. Only the rupture ice before the noticeable bulges displays aggregation behavior. The area of the aggregation ice can be calculated after estimating the width of the aggregation ice from the pictures, because the joints are two-dimensional surfaces. The accumulated width of the aggregation ice can be measured in the picture. Then the aggregation area of the rupture ice can be calculated as

$$A_i = \frac{\sum_{k=1}^{n} L_k}{L_{\text{joint}}} \times 100\% \tag{1}$$

where $L_k$ is the width of the aggregation ice for the bulge $k$. $L_{joint}$ =10 cm which is the trace length of the joint.

This point has been added in our revised paper. (Details can be found on

The data points of aggregation area of rupture ice are not the mean values of three samples. We are very sorry about the negligence of our previous experimental design and the limitation of the experiment condition, so we only tracked and monitored the surface characteristics of only one group of ice-filled samples instead of all the samples from three parallel tests. We monitored all the data of samples in the third group of parallel experiments. Although only one set of data is monitored, the experiment data are reliable and the experiment phenomenon is remarkable. We will pay attention to these details in future studies.

10. L194, provide references.

Response: Thank you for your good suggestion. Lou et al. (2022) claimed that plain ice has strong brittleness at the temperature from -20 ℃ to -5 ℃. The brittle-ductile transition interval of pure ice is not clear according to the previous literature, so we take the characteristics of the failure surface as the basis for judging ductile failure and brittle failure. Rupture ice (macroscopic failure phenomena) will be produced under brittle failure condition. In the ice-filled joint, these parts of the rupture ice cannot squeeze out from the joint and the aggregation of rupture ice along the noticeable bulges will be created. The increasing aggregation area of the rupture ice in Fig. 6 further proves that the brittleness of ice increases with decreasing the freezing temperature. The maximum shear

displacement before failure is smaller at -15 ℃. The related reference and explanation are added in our revised paper. (Details can be found in Lines 214 to 219, which are marked in red.)

11. In my opinion, the shear rate range is quite small. This is why the shear strength experiences neglectable change. The explanations in L242-254 are less convincing. Therefore, please give the exact shear rate adopted in the references when discussing the ductile to brittle transitions.

Response: According to the research of Mamot et al. (2018) and Fukuzawa et al. (1993), the brittle-ductile transition of ice under the shear process occurs around the strain rate of $10^{-4}$ $s^{-1}$~$10^{-3}$ $s^{-1}$. In this study, the shear displacement rate is from 0.2 mm/min to 0.8 mm/min, corresponding to the strain rates from $1.67 \times 10^{-3}$ $s^{-1}$ to $6.67 \times 10^{-3}$ $s^{-1}$. Therefore, the shear rate in this study is very close to the threshold of brittle-ductile transition given in the previous literature. Figure 12 shows that a high shear rate will induce brittle failure of joint ice and more fragmented ice crystals are produced. As a result, the shear strength reduces with increasing shear rates from 0.2 mm/min to 0.8 mm/min. In this study, the exact shear rate for the brittle and ductile transition of ice-filled joints is not accurately determined due to the limitation of the shear rate range. More further shear experiments should be carried out on the ice and ice-filled joints by adopting a larger range of the shear rate. The exact shear rate adopted in the references is added in our revised

paper. In addition, some new discussions also are added. (Details can be found in Lines 273 to 284, which are marked in red.)

12. L267-271, does it mean that the joint surface was not fully filled by ice when d=2mm?

Response: The joint surface was fully filled by ice when $d$=2mm. We are sorry for your misunderstanding due to our improper expression. The standard roughness curve is proposed by Bardon and Choubey (1977), and the maximum height difference of the standard JRC curves is measured by importing the standard JRC profiles into AutoCAD. The maximum height difference is defined between one JRC curve. The joint opening is the vertical distance between the upper and lower blocks. Therefore, the maximum height difference and joint opening are different. The comparison between them are given in the following figure. The definition of the joint opening is added in our revised paper. (Details can be found on Line 297~298, which are marked in red.)

[Figure]

**Figure** comparison between the maximum height difference and the joint opening

13. In Figs. 13 and 14, it is better to adopt the joint opening as the horizontal axis for evaluating the effect of joint opening.

Response: Thank you for your suggestion. Indeed, it is better to adopt the joint opening as the horizontal axis for evaluating the effect of joint opening if only contrast with different joint openings. However, the horizontal axis of joint number aims to highlight at which joint opening the shear strength and shearing dilatancy of all standard joints is similar to that of solid ice. Therefore, a line for the shear strength of solid ice is plotted in these figures. In Fig. 14, we can observe that the shear strength of ice-filling joints with infilling thickness of 14 mm is equal to that of solid ice (approximately 0.83 MPa). In Fig. 16, when the infilling thickness is 14 mm, the shear dilatancy of the ice-filling joints is close to the pure ice (0.224 mm). If you think there is something wrong, please let us know. Thank you for your good suggestions.

---

## Author Comment (AC3)

Nov 20, 2022

Dear referees

Thanks for your comments and suggestions. Based on these comments and suggestions, we have made careful modifications on this manuscript. Appended to this letter is our point-to-point response to the comments. These comments were reproduced and our response were given directly afterward in a different color (blue).

We hope that the manuscript can be accepted for publication in **The Cryosphere**. If you have any question, please contact us immediately. We are grateful for your attention to our manuscript. Once again, thanks very much for your arduous work and instructive suggestions to our manuscript processing.

Sincerely,

Dr. Shibing Huang
* * *
School of Resources and Environmental Engineering,

Wuhan University of Science and Technology,

Wuhan, 430081, P.R. China

Telephone: +86 185-0275-5916

E-mail: huangshibing@wust.edu.cn

Response to the comments

Response to referee:

1. Abstract part should be compressed and polished. Line 24~25 should be deleted, because it is not the main conclusion of this study.

Response: Thank you for your good suggestion. We have deleted L24-25 and compressed the abstract. In addition, more important results have been added in Lines 15 to 22, mainly including the effect of joint roughness, freezing temperature, joint opening and normal stress. (Details can be found in the abstract in our revised paper, which are marked in red.)

2. Red sandstone blocks with dimensions of 100×100×50 mm were used to engrave roughness curve. However, a certain amount of specimen thickness is consumed during engraving. The ice-filled joint masses may not be a standard 100×100×100 mm cube. This point should be claimed and whether the height of ice-filling samples has any effect on the shear strength should be explained.

Response: Thank you for your suggestion, the standard red sandstone block after engraving roughness curve will have different heights, so we added a steel gasket to ensure the ice-filled joint blocks have a unified height during the shear experiment. We ensured the block's height error was within 0.5 %, which makes the same joint opening under the same experimental condition. The location of the steel sheet is shown in the

following figure. This point has been clarified in our revised paper (Details can be found on Line 136~137 and in Fig. 4 in our revised paper, which are marked in red.)

[Figure]

3. Line 69. "normal stress" should be deleted, because the previous literature has considered the effect of normal stress

Response: Thank you for your good suggestion. We rechecked the previous literature and found that it had considered the effect of normal pressures, so we deleted "normal stress" in Line 68. (Details can be found in Lines 68 to 69, which are marked in red.)

4. In Fig. 2, it can be seen that when studying the influence of different shear rates and joint openings on the shear strength of ice-filled joint, the experimental temperature is -5 ℃. Why -15 ℃ is used as the experimental temperature under different normal stress. This may be not conducive for comparison. Can you explain it?

Response: In the normal stress group, we want to investigate the influence of the normal stress on the shear strength of the ice-filling joints.

The freezing temperature, shear rate and joint opening are the same. Therefore, this group is not related with the freezing temperature. In addition, the difference of the shear strength between -15 ℃ and -5 ℃ is small. The change rule of the shear strength against the normal stress are similar no matter which freezing temperature is adopted. Therefore, the freezing temperature has no effect on the change rule of shear strength caused by normal stress.

5. In Fig. 7, why the error bars are not added, because three parallel experiments were conducted?

Response: We do not give the uncertainty, because we only continuously capture the shear rupture picture for one group. Although only one set of data is monitored, the experiment data are reliable and the experiment phenomenon is remarkable. In order to make the experimental results more convincing, we will pay attention to these details in the future studies.

6. Line 193~195. The turning point of brittle and ductile failure of pure ice at different temperatures is not clear. It is suggested to compare the present results with other studies and describe the macroscopic or microscopic failure phenomena more clearly.

Response: Thank you for your good suggestion. Lou et al. (2022) claimed that plain ice has strong brittleness at the temperature from -20 ℃ to -5 ℃. The brittle-ductile transition interval of pure ice is not clear

according to the previous literature, so we take the characteristics of the failure plane as the basis for judging ductile failure and brittle failure. Rupture ice (macroscopic failure phenomena) will be produced under brittle failure condition. In the ice-filled joint, these parts of the rupture ice cannot squeeze out the joint and will form an aggregation of rupture ice area along the noticeable bulges. The increasing aggregation area of the rupture ice in Fig. 6 further proves that the brittleness of ice increases with decreasing the freezing temperature. The maximum shear displacement before failure is smaller at -15 ℃. The above comparisons are added in our revised paper. Thank you for your instructive comments. (Details can be found in Lines 214 to 219, which are marked in red.)

7. Figure 16 - This figure show the effect of normal stress on the peak shear strength of ice-filled joints, the experimental condition should be corrected as T = -15 ℃, v = 0.2 mm/min and d = 2 mm.

Response: Thank you for your good suggestion. We have replaced " $\sigma_n =$ 0.5 MPa" as "$d = 2mm$" in Fig. 17. (Details can be found in Lines 381 to 382, which are marked in red.)

8. It is interesting to propose the noticeable bulges to explain the change of shear strength of ice-filling joints against the joint roughness. However, how to determine the noticeable bulges and what is the characteristic of these noticeable bulges need more evidence.

Response: Thank you for your good suggestion. The "noticeable bulges"

are pointed out in Fig. 7. The bulges causing the reduction of joint width and aggregation of ice are called noticeable bulges. The noticeable bulges have larger inclination angles and they are far away from the joint edges. The definition of the noticeable bulge is added in our revised paper. (Details can be found on Line 206~208 in our revised paper, which are marked in red.)

9. The conclusions should be compressed and improved, for example the sentence "Above all, this study … normal stress" may be shorten or deleted.

Response: Thank you for your good suggestion. We have deleted the sentence "Above all, this study … normal stress" and polished the conclusions. (Details can be found in conclusion in our revised paper, which are marked in red.)

10. Line 329 to 333 – Where should be replaced by "where".

Response: Thank you for your good suggestion. We have replaced "Where" as "where". (Details can be found in Line 363, which are marked in red.)

11. Line 12. "was" should be replaced by "were". It is suggested to polish the English carefully.

Response: Thank you for your good suggestion. We have replaced "was" with "were" and done a double check about the English and fixed grammatical errors. (Details can be found in Line 12, which are marked

in red.)

12. It is suggested that the authors search for the literature related to "ice-filled rock joints" or "ice-filled rock flaw" and cite it appropriately in the introduction.

Response: Thank you for your good suggestion. We have added the literature related to "ice-filled rock joints" or "ice-filled rock flaw" in our revised paper. (Details can be found on Line 36~40, which are marked in red.)